# Isoprostanoid Profiling of Marine Microalgae

**DOI:** 10.3390/biom10071073

**Published:** 2020-07-18

**Authors:** Claire Vigor, Camille Oger, Guillaume Reversat, Amandine Rocher, Bingqing Zhou, Amandyne Linares-Maurizi, Alexandre Guy, Valérie Bultel-Poncé, Jean-Marie Galano, Joseph Vercauteren, Thierry Durand, Philippe Potin, Thierry Tonon, Catherine Leblanc

**Affiliations:** 1Institut des Biomolécules Max Mousseron, IBMM, Université de Montpellier, CNRS, ENSCM, Faculté de Pharmacie, 34093 CEDEX 5 Montpellier, France; camille.oger@umontpellier.fr (C.O.); guillaume.reversat@umontpellier.fr (G.R.); Amandine.rocher@umontpellier.fr (A.R.); bingqing.zhou@umontpellier.fr (B.Z.); amandyne.linares-maurizi@umontpellier.fr (A.L.-M.); alexandre.guy@umontpellier.fr (A.G.); valerie.bultel@umontpellier.fr (V.B.-P.); jean-marie.galano@umontpellier.fr (J.-M.G.); joseph.vercauteren@umontpellier.fr (J.V.); thierry.durand@umontpellier.fr (T.D.); 2Integrative Biology of Marine Models, LBI2M (Sorbonne Université/CNRS), Station Biologique de Roscoff (SBR), 29680 Roscoff, France; philippe.potin@sb-roscoff.fr (P.P.); thierry.tonon@york.ac.uk (T.T.); catherine.leblanc@sb-roscoff.fr (C.L.); 3Centre for Novel Agricultural Products, Department of Biology, University of York, Heslington, YO10 5DD York, UK

**Keywords:** microalgae, PUFAs, isoprostanoids, oxidative stress, micro-LC-MS/MS

## Abstract

Algae result from a complex evolutionary history that shapes their metabolic network. For example, these organisms can synthesize different polyunsaturated fatty acids, such as those found in land plants and oily fish. Due to the presence of numerous double-bonds, such molecules can be oxidized nonenzymatically, and this results in the biosynthesis of high-value bioactive metabolites named isoprostanoids. So far, there have been only a few studies reporting isoprostanoid productions in algae. To fill this gap, the current investigation aimed at profiling isoprostanoids by liquid chromatography -mass spectrometry/mass spectrometry (LC-MS/MS) in four marine microalgae. A good correlation was observed between the most abundant polyunsaturated fatty acids (PUFAs) produced by the investigated microalgal species and their isoprostanoid profiles. No significant variations in the content of oxidized derivatives were observed for *Rhodomonas salina* and *Chaetoceros gracilis* under copper stress, whereas increases in the production of C18-, C20- and C22-derived isoprostanoids were monitored in *Tisochrysis lutea* and *Phaeodactylum tricornutum*. In the presence of hydrogen peroxide, no significant changes were observed for *C. gracilis* and for *T. lutea*, while variations were monitored for the other two algae. This study paves the way to further studying the physiological roles of isoprostanoids in marine microalgae and exploring these organisms as bioresources for isoprostanoid production.

## 1. Introduction

Marine ecosystems account for approximately half of the global primary production, and unicellular eukaryotes, e.g., photosynthetic microalgae, as part of the phytoplankton, are major contributors to this ocean productivity [1]. These organisms also play critical roles in the biogeochemical cycle of many chemical elements, including carbon, nitrogen, sulfur, phosphorus and silica. Currently, more than 35,000 species of microalgae have been described, which likely represent only a small part of the supposed biodiversity, since their number of species has been estimated to range between 200,000 and 800,000 [2].

Microalgae can grow mostly autotrophically but, also, heterotrophically or mixotrophically according to culture conditions and metabolic capacities. These are related to the different environments inhabited by these organisms, as well as their evolutionary history that shaped their network of biochemical pathways [3]. Some microalgae exhibit high contents in proteins, lipids, sugars and pigments, making them attractive for a number of biotechnological applications. Such potential has been investigated for the bio-based production of a wide range of compounds for the food, feed, energy, agriculture and health sectors [4].

Among the interesting compounds produced by microalgae are the omega-3 (ω-3 or n-3) long-chain polyunsaturated fatty acids (n-3 PUFAs), eicosapentaenoic acid (EPA; C20: 5n-3) and docosahexaenoic acid (DHA; C22:6n-3). While land plants and microalgae can produce the medium-chain α-linolenic acid (ALA; C18:3n-3), only the latter organisms can convert this precursor into EPA and DHA. These three fatty acids are considered as essential in human nutrition, because ALA cannot be synthesized de novo by humans, and the metabolic conversion efficiency of dietary ALA into EPA and DHA is low and insufficient to meet physiological demands [5]. These n-3 PUFAs have been shown to provide significant benefits on human health [6,7], notably in mitigating a number of pathological conditions, including cardiac diseases [8]. They are also important for the healthy development of the neural system [9,10], and as such, they are necessarily included in infant formula. Very recently, it has been proposed that dietary n-3 PUFAs selectively drive the expansion of adipocyte numbers to produce new fat cells and store saturated fatty acids, enabling the homeostasis of healthy fat tissue [11]. At present, marine fishes and fish oils are the main commercial sources of n-3 PUFAs. However, the suitability of these sources of PUFAs for human consumption has been questioned, notably because of biosafety (e.g., contents in heavy metals) and of overfishing. In addition, the current supply of n-3 PUFAs from these traditional sources is insufficient to satisfy human nutritional requirements [12]. Therefore, new sources of n-3 PUFAs have been investigated, such as wild-type and engineered microbes, including microalgae [13], and the extraction of fish oil from genetically modified crops [14].

It is well-established that PUFAs are highly reactive species sensitive to oxidation because of the presence of bis-allylic structures, in which α-hydrogen atoms are easily removed by the action of free radicals. Some of these free radicals, named reactive oxygen species, are produced under oxidative stress (OS) conditions and react with PUFAs to form, spontaneously through enzymatic reactions, oxidized derivatives of PUFAs. All of these oxidized metabolites are grouped under the term oxylipins. Most of the oxylipins studied so far are derived from the enzymatic transformation catalyzed by enzymes such as lipoxygenases or dioxygenases. During the last two decades, it has been shown that the nonenzymatic oxidation of PUFAs (NEO-PUFAs) leads to other valuable compounds. ALA are precursors of phytoprostanes (PhytoPs), arachidonic acid (AA; C20: 4n-6) of isoprostanes from the serie 2 (IsoPs, serie 2), EPA of isoprostanes from the serie 3 (IsoPs, serie 3), AdA (docosatetraenoic acid; C22: 4n-6) of dihomo-isoprostanes and dihomo-isofurans, and DPA_n-6_ (docosapentaenoic acid; C22: 5n-6) and DHA of neuroprostanes (NeuroPs) (Figure 1 and Figure 2). These NEO-PUFAs are considered to be very good markers of OS in plants and animals. They have also been shown to act as lipid mediators, with key functions in various cell-signaling pathways [15], and have been suggested to be potentially beneficial for human health [16].

Recently [17], we have investigated variations in the isoprostanoid contents of red and brown macroalgae after exposure to oxidative (heavy metal) stress conditions [18]. In addition, changes in the production of isoprostanoids derived from C18, C20 and C22 fatty acids were observed in the microalga *Phaeodatylum tricornutum* subjected to oxidative stress by cultivation under increasing doses of hydrogen peroxide (H_2_O_2_) [19]. This work suggested that nonenzymatic oxylipins in *P. tricornutum* may be involved in the control of important processes under various physiological and environmental conditions. In view of these findings, and to go further in the study of the potential production of NEO-PUFAs by algae, we were first interested in increasing our knowledge on the distribution of NEO-PUFAs in different lineages of marine microalgae by establishing qualitative and quantitative profiles under laboratory culture growth conditions. Based on previous analyses of fatty acids and lipid compositions in marine microalgae and, notably, the high production of EPA and DHA by some of them [20,21,22], we decided to select the four following species: the diatoms *Phaeodactylum tricornutum* and *Chaetoceros gracilis* known to exhibit high content of EPA, the haptophyte *Tisochrysis lutea* that has been shown to produce elevated amounts of ALA and DHA and the cryptophyte *Rhodomonas salina* that harbors similar and high levels of EPA and DHA. Our second objective was to assess changes in the isoprostanoid profiles of the selected microalgae under altered physiological conditions in relationship with the exposure to oxidative stress (copper and hydrogen peroxide treatments).

## 2. Materials and Methods

### 2.1. Chemicals and Reagents

All the NEO-PUFA analytical standards, as well as the internal standard (IS) mixture (C19-16-F_1t_-PhytoP and C21-15-F_2t_-IsoP) used to determine the calibration curve ratio, were synthesized according to previously described procedures [23,24,25,26]. NEO-PUFA analytical standards were as follows: phytoprostanes (9-L_1_-PhytoP, *ent*-9-L_1_-PhytoP, *ent*-16-*epi*-16-F_1t_-PhytoP, 9-F_1t_-PhytoP, 16-F_1t_-PhytoP + 9-*epi*-9-F_1t_-PhytoP, 16(*RS*)-16-A_1_-PhytoP, 16-B_1_-PhytoP and *ent*-16-B_1_-PhytoP); phytofurans (*ent*-16(*RS*)-9-*epi*-ST-Δ^14^-10-PhytoF, *ent*-9(*RS*)-12-*epi*-ST-Δ^10^-13-PhytoF and *ent*-16(*RS*)-13-*epi*-ST-Δ^14^-9-PhytoF) coming from oxidation of the C18 n-3 ALA; isoprostanes derived from the C20 n-6 AA (15-F_2t_-IsoP, 15-*epi*-15-F_2t_-IsoP, 5-F_2t_-IsoP, 5-*epi*-5-F_2t_-IsoP and 5-F_2c_-IsoP); isoprostanes coming from the oxidation of C20 n-3 EPA (8-F_3t_-IsoP, 8-*epi*-8-F_3t_-IsoP, 18-F_3t_-IsoP and 18-*epi*-18-F_3t_-IsoP); dihomo-isoprostanes and dihomo-isofurans derived from the C22 n-6 AdA (*ent*-7(*RS*)-7-F_2t_-dihomo-IsoP and 7(*RS*)-ST-Δ^8^-11-dihomo-IsoF); neuroprostanes coming from the oxidation of C22 n-3 DHA (4(*RS*)-4-F_4t_-NeuroP, 10-F_4t_-NeuroP, 10-*epi*-10-F_4t_-NeuroP, 20-F_4t_-NeuroP and 20-*epi*-20-F_4t_-NeuroP) and those derived from the oxidation of C22 DPA_n-6_ (4(*RS*)-4-F_3t_-NeuroP_DPAn-6_). The only exception is 16 (*RS*)-16-A_1_-PhytoP that were purchased from Cayman Chemicals (Ann Arbor, MI, USA). Liquid chromatography – mass spectrometry (LC-MS) grade water, methanol, acetonitrile and chloroform were obtained from Fisher Scientific (Loughborough, UK). Hexane (CHROMASOLV for high performance liquid chromatography - HPLC), formic and acetic acid, ammonia and potassium hydroxide (Fluka for mass spectrometry) were purchased from Sigma-Aldrich (Saint Quentin Fallavier, France). Ethyl acetate (HPLC grade) was acquired from VWR (Fontenay-sous-bois, France). Solid-phase extraction (SPE) cartridges Oasis MAX with mixed polymer phase (3 mL, 60 mg) were obtained from Waters (Milford, MA, USA).

### 2.2. Microalgal Species

The four microalgae (*Tisochrysis lutea* RCC 1349, *Phaeodactylum tricornutum* RCC 69, *Chaetoceros gracilis* and *Rhodomona salina* RCC 20) used in this study were obtained from the Roscoff Culture Collection (RCC) and from the EMBRC Roscoff culture facilities for *C. gracilis*. This latter strain is a kind gift from the Experimental Mollusc Hatchery of Ifremer at Argenton (France) and is cultivated for larvae feeding (Robert et al., 2004, https://archimer.ifremer.fr/doc/2004/rapport-1546.pdf).

### 2.3. Cultivation of Microalgae and Oxidative Stress Treatments

Microalgae were grown in Conway medium [27], commonly used in aquaculture, at a temperature of 18 °C and under a continuous light intensity of 300 μmoles m^−2^ s^−1^ for biomass production [28,29]. The volume of the culture was gradually brought, by successive subculture in increasing volumes of the medium, to a final volume of 10 L in Nalgene flasks placed under constant aeration. Cells were harvested after reaching the stationary phase. After centrifugation (5000 rpm for 25 min), the supernatant was removed, and the pellet resuspended in 2 L of 0.45 μm filtered and autoclaved natural seawater (FSW) collected offshore at Roscoff (at a site with no direct chemical influence from the shore) and free of organic matter. After agitation to ease resuspension, cells were spun down again for 25 min at 5000 rpm, then resuspended as described above in a final volume of 900 mL. This suspension was split into nine glass flasks previously washed overnight with 1% HCl to limit the Cu adsorption and rinsed with mqH_2_O and FSW. The 100 mL cell volume was brought to 1 L with FSW. Three flasks were considered as the control, three were used for copper stress (Cu(II) as CuCl_2_) and three for incubation in the presence of hydrogen peroxide (H_2_O_2_).

Oxidative stress was triggered by adding Cu(II) as CuCl_2_ (Merck, Germany) at a final concentration of 0.3 μM or H_2_O_2_ at 1 mM. After 24 h of incubation under the conditions described above, cells were harvested as explained in the previous section. Supernatants were discarded, and cells were washed one time using FWS and centrifugated before freezing in liquid nitrogen and freeze-drying. Algal material was stored at −20 °C until analysis.

### 2.4. Preparation of Algal Samples for Lipidomic Analysis

During the preparation of samples for such analysis, we made two important observations. First, we noticed that one sample of *P. tricornutum* obtained under H_2_O_2_ stress condition contained some water after lyophilization. This sample was not considered further for extraction. In addition, one sample of *T. lutea* obtained after copper stress showed significant differences in color and texture during the extractive process when compared to the other samples. The data acquired in LC-MS/MS for this latter sample showed numerous outliers (Grubbs’ statistical test; data not shown), which were discarded for subsequent analysis.

A protocol similar to what was described for our previous work on macroalgae was applied for lipidomic analysis [18]. Freeze-dried microalgal samples were coarsely reduced to powder using, first, a Mixer Mill MM400 (Retsch^®^) bench top unit. Then 100 mg of powder was added in grinding matrix tubes (lysing matrix D, MP Biochemicals, Illkirch, France) with 25 μL of BHT (butylated hydroxytoluene 1% in water) and 1 mL of MeOH. Tubes were placed in a FastPrep-24 (MP Biochemicals), and samples were ground for 30 s at a speed of 6.5 m/s. Suspensions were transferred into a 15 mL centrifuge tube, and 1 mL of MeOH, 4 μL of IS (1 ng/μL) and 1.5 mL of phosphate buffer (50 mM, pH 2.1, prepared with NaH_2_PO_4_ and H_3_PO_4_) saturated in NaCl were added. Tubes were then stirred for 1 h at 20 °C. Subsequently, the mixture was vortexed and centrifuged at 5000 rpm for 5 min at room temperature. The organic phase was recovered in Pyrex tubes, and the solvent was dried under a stream of nitrogen at 40 °C. Afterward, lipids were hydrolyzed with 950 μL of KOH for 30 min at 40 °C. After incubation, 1 mL of formic acid (FA; 40 mM, pH 4.6) was added before running the SPE separation. First, SPE Oasis MAX cartridges were conditioned with 2 mL of MeOH and equilibrated with 2 mL of formic acid (20 mM, pH 4.5). After loading the sample, the cartridges were successively washed with 2 mL of NH_3_ (2% (*v*/*v*)), 2 mL of a mixture of formic acid (20 mM):MeOH (70:30, *v*/*v*), 2 mL of hexane and, finally, 2 mL of a hexane:ethanol:acetic acid (70:29.4:0.6, *v*/*v*/*v*) mixture. Lastly, all samples were evaporated to dryness under a nitrogen flow at 40 °C for 30 min and reconstituted in 100 μL of a mobile phase (solvent A: water with 0.1% (*v*/*v*) of formic acid, solvent B: ACN:MeOH, 8:2 *v*/*v* with 0.1% (*v*/*v*) of formic acid and A:B ratio 83:17) for injection.

### 2.5. Preparation of Samples for Analysis of Extraction Yield and Matrix Effect

Parameters related to extraction yield (EY) and matrix effect (ME) were determined for a better description of microalgal isoprostanoid profiles. To this aim, three sets of samples were prepared. The first one was obtained by addition of 6.4 µL of two different concentrations of a standard mixture (36 PUFA oxidized metabolites at 0.5 and 8 ng/mL) into 100 mg of freeze-dried microalgae at the beginning of the extraction process described above to reach concentrations of 32 and 512 ng/g, respectively. This corresponds to the “pre-spiked samples”. For the second set of samples, extraction was done as explained in the previous section on 100 mg of microalgae up to the elution step. Then, eluates were spiked with 6.4 µL of the two different concentrations of standard mixture used for the “pre-spiked samples” and processed to complete the algal sample preparation protocol. These samples were named “post-spiked samples”. The third set of samples consisted of standard solutions (final concentrations of 32 and 256 ng/mL) prepared in 100 μL of the mobile phase H_2_O: ACN: FA (83: 17: 0.1, *v*/*v*/*v*). All sets of samples were analyzed using the LC-MS/MS system described below. The EY was calculated as the percentage difference between peaks areas of standards in pre-spiked and post-spiked samples. The ME was determined as the percentage difference between peak areas of standard added to the extracted samples (post-spiked sample) and pure standards diluted into the mobile phase. The ME and EY were calculated for each isoprostanoid and for each species.

### 2.6. Micro-LC-MS/MS Analysis

All LC-MS analyses were carried out using an Eksigent^®^ MicroLC 200 Plus (Eksigent Technologies, CA, USA) on a HALO C18 analytical column (100 * 0.5 mm, 2.7 μm; Eksigent Technologies, CA, USA) kept at 40 °C. The mobile phase consisted of a binary gradient of solvent A (water with 0.1% (*v*/*v*) of formic acid) and solvent B (ACN:MeOH, 8: 2, *v*/*v* with 0.1% (*v*/*v*) formic acid). The elution was performed at a flow rate of 0.03 mL/min using the following gradient profile (min/%B): 0/17, 1.6/17, 2.85/21, 7.3/25, 8.8/28.5, 11/33.3, 15/40, 16.5/95 and 18.9/95 and then returned to the initial conditions. Under these conditions, no sample contamination or sample-to-sample carryover was observed.

Mass spectrometry analyses were performed on an AB SCIEX QTRAP 5500 (Sciex Applied Biosystems, ON, Canada). The ionization source was electrospray (ESI), and it was operated in the negative mode. The source voltage was kept at −4.5 kV, and N_2_ was used as the curtain gas. The multiple ion monitoring (MRM) of each compound was predetermined by MS/MS analysis to define the two transitions for quantification (T_1_) and specification (T_2_) (Appendix A). The analysis was conducted by monitoring the precursor ion to the product ion (T_1_). Peak detection, integration and quantitative analysis were performed by MultiQuant 3.0 software (Sciex Applied Biosystems). The quantification of the isoprostanoids was based on calibration curves obtained from the analyte to the IS area under the curve ratio. Linear regression of six concentrations of standards mixture (16, 32, 64, 128, 256 and 512 pg/μL) of each standard were calculated. The sensitivity of the method was evaluated through limit of detection (LOD) and limit of quantification (LOQ) parameters, which were defined as the lowest concentration with a signal to noise ratio above 3 and 10, respectively.

### 2.7. Statistical Analysis

All statistical analyses were performed with R [30], all graphics were created with different functions of the tidyverse package [31] and all the tables with kableExtra package [32]. Analyte concentrations were compared by one-way analysis of variance (ANOVA) and post-hoc (Tukey) test for multiple comparison using rstatix package [33]. For all analyses, the significance threshold was 0.05 for the p-value resulting from the statistical test used.

## 3. Results

### 3.1. Analysis of Extraction Yield and Matrix Effect

The analysis of NEO-PUFAs in natural matrices is extremely challenging, requiring highly sensitive and specific methods for their profiling and characterization. Therefore, a protocol relying on the specific extraction of lipophilic compounds (Folch’s extraction), combined with a step of SPE to eliminate potentially interfering substances, was implemented to obtain an extract enriched in NEO-PUFAs. Such a protocol has proven to be efficient for similar analyses in the past [34,35]. Isoprostanoids were subsequently separated, identified and quantified using a micro-LC-MS/MS method validated by previous studies [36,37,38]. Identification relied on retention times observed during spiked experiments, determination of molecular masses and the analysis of specific MS/MS transitions. Calibration curves for the calculation of the concentrations were established for 32 compounds (Appendix A), as well as LODs and LOQs. Values were found to be dependent of the type of isoprostanoids and ranged between 0.16 and 0.63 pg injected for the LODs and between 0.16 and 1.25 pg injected for the LOQs. In addition, based on previous experiments done on macroalgae (Vigor et al., 2018), we decided to assess the influence of the matrix effect (ME) on the extraction protocol, since this can affect the extraction yields (EY) and/or mass ionization. Therefore, algal samples spiked with two different concentrations of a standard mixture (SM_32_ or SM_256_) were analyzed to calculate the EY and the ME, which subsequently enabled the determination of the efficiency of the sample processing (Appendix A). The extraction yield, a parameter specific to each compound (standards and IS), allowed the evaluation of product losses that could happen by retention on the SPE cartridge and/or by partial elution during the washing steps. For the majority of analytes of *C. gracilis* and *R. salina*, the apparent loss of compounds during SPE was between 10% and 20%. The results were most often similar for the two spiked concentrations (32 and 512 ng/g). Regarding the type of compounds (PhytoPs, PhytoFs, IsoPs or NeuroPs), no specific trend was noticed. As far as *P. tricornutum* and *T. lutea*, the calculated extraction yield was more than 100% for some analytes, corresponding probably to the coelution of a compound that presents the same MRM transition. Note in the table the values of two or even three units considered to be outliers. To complete this validation, the matrix effect, corresponding to an ion-suppression/enhancement of coeluted matrix compounds, was evaluated. As for EY, ME is specific to each isoprostanoid, and there was no similar behavior across the same class of compounds or across selected species.

For the sake of clarity, results are presented species-by-species in the next sections. In addition, Table 1 provides a summary of the relative percentage distribution of each type of isoprostanoid (ALA, AA and EPA; AdA, EPA, DPA and DHA) in the four species studied.

### 3.2. Rhodomonas Salina

Analysis of the isoprostanoid profile of this species revealed the presence of 35 isoprostanoids (Table 2 and Appendix A). The concentrations of metabolites were comprised between 13.4 ng/g for the epimers 4(*RS*)-4-F_3t_-NeuroP and 2 µg/g for 16-B_1_-PhytoP. The total amount of isoprostanoids in *R. salina* was 10.6 µg/g.

Considering the four PhytoPs corresponding to epimers 9-F_1t_-PhytoP, 9-*epi*-9-F_1t_-PhytoP, 16-F_1t_-PhytoP and 16-*epi*-16-F_1t_-PhytoP, plus two other derivatives (16-B_1_-PhytoP and 9-L_1_-PhytoP) and two pairs of phytofuranoid forms (*ent*-16(*RS*)-9-*epi*-ST-Δ^14^-10-PhytoF and *ent*-9(*RS*)-12-*epi*-ST-Δ^10^-13-PhytoF), ALA is considered as the main source of isoprostanoids in *R. salina*. This is confirmed when assessing the amounts of oxidized derivatives produced for each potential precursor. Those from ALA represent an average value of 7.6 µg/g of algal dry weight, with the 16-B_1_-PhytoP and the *ent*-16(*RS*)-9-*epi*-ST-Δ^14^-10-PhytoF being the most abundant (2 µg/g and 1.8 µg/g, respectively). DHA was also inferred to produce a wide range of compounds, with up to ten stereoisomeric NeuroPs that could be arranged by pairs. The sum of the DHA derivatives was 0.8 µg/g, i.e., approximately ten times less than the sum of the ALA derivatives. *R. salina* also synthesized six EPA derivatives, again as epimers that could be classified by pairs, which corresponded to an amount of 1.9 µg/g. Therefore, compared to the DHA-derived isoprostanoids, the EPA products had slightly less structural diversity but accumulated at a higher content. Among the other molecular species of interest, it is worth mentioning those derived from AA: five representatives (15-*epi*-15-F_2t_-IsoP, 15-F_2t_-IsoP, 5(*RS*)-5-F_2t_-IsoP and 5-F_2c_-IsoP) were identified, for a total content of 0.26 µg/g. To complete this description, other isoprostanoids were observed, including AdA derivatives (7(*RS*)-ST-Δ^8^-11-dihomo-IsoF at the level of 0.06 µg/g) and DPA_n-6_ derivatives (4(*RS*)-4-F_3t_-NeuroP_DPAn-6_ at the level of 0.01 µg/g). Based on this analysis, it is interesting to note that, while the cryptophyte *R. salina* is known to produce high amounts of EPA and DHA [39,40,41], the most abundant isoprostanoids were derived from the C18 ALA (71.5 % of the total amount of PUFA-oxidized derivatives).

After oxidative stress, no modification in the diversity of the molecules identified could be noticed. All the compounds observed under the control condition were still present under the OS condition, regardless of the type of stress applied. Few significant changes were observed in the content of the 35 NEO-PUFAs measured initially. In fact, based on the statistical analysis, the amount of only one compound showed a significant increase (Appendix A) between the control condition and H_2_O_2_ stress. Indeed, it was observed that the content of the two 7(*RS*)-ST-Δ^8^-11-dihomo-IsoF epimers doubled, from 0.06 µg/g to 0.13 µg/g, after OS (Table 1). In addition, when comparing profiles obtained after copper and peroxide hydrogen additions, differences were noted for three compounds: 8-F_3t_-IsoP, *ent*-16(*RS*)-9-*epi*-ST-Δ^14^-10-PhytoF and 7(*RS*)-ST-Δ^8^-11-dihomo-IsoF (Figure 3).

### 3.3. Tisochrysis Lutea

*T. lutea* has a greater diversity of compounds identified compared to the other species investigated, with 38 isoprostanoids observed. This consisted of 16 derivatives coming from ALA oxidation, ten derivatives from DHA, three from EPA, three from AA, four from AdA and, finally, two from DPA_n-6_. Despite this increased diversity, the total amount of isoprostanoids measured in this haptophyte was lower than what was measured in the cryptophyte *R. salina*, i.e., 7 µg/g. The details of the oxidized PUFA derivatives grouped by family are as follow: 4.8 µg/g from ALA, 1.9 µg/g from DHA, 0.008 µg/g from EPA, 0.012 µg/g from AA, 0.045 µg/g from AdA and 0.135 µg/g from DPA_n-6_. The levels of individual metabolites were comprised between 1.24 ng/g for 8-*epi*-8-F_3t_-IsoP and 0.988 µg/g for 16-B_1_-PhytoP, which represents a 1000-fold difference and indicated that isoprostanoids can be produced at very different ranges in this microalga species (Table 3).

The production of these molecules is in accordance with the fatty acid profile of *T. lutea*, since ALA and DHA are the most abundant PUFAs measured in this alga [39,42], and the contents of their oxidized derivatives represented 97% of the total amount of identified isoprostanoids. Taking a closer look at the four main families of metabolites, the amounts of ALA derivatives ranged from 0.07 µg/g (*ent*-9-D_1t_-PhytoP) to 1 µg/g (16-B_1_-PhytoP) and those of AA derivatives from 5 ng/g (5(*RS*)-5-F_2t_-IsoP) to 6 ng/g (5-F_2c_-IsoP). The contents of the EPA derivatives went from 1.2 ng/g for the lowest (8-*epi*-8-F_3t_-IsoP) to 4.5 ng/g for the highest (18-*epi*-18-F_3t_-IsoP) and, for DHA derivatives, from 0.1 µg/g for the lowest (10-*epi*-10-F_4t_-NeuroP) to 0.5 µg/g for the highest (4(*RS*)-4-F_4t_-NeuroP).

When assessing the impact of oxidative conditions, the oxidized metabolite diversity remained unchanged. Furthermore, no variation in the isoprostanoid content of *T. lutea* was monitored after incubation in the presence of H_2_O_2_. In contrast, the copper treatment had a strong effect; the contents of 17 among the 38 oxidized derivatives increased under this stress condition. Two compounds were very significantly impacted, as shown by the p-values adjusted for multiples comparisons: 5-F_2c_-IsoP (*p* < 0.0005) and *ent*-16-*epi*-16-F_1t_-PhytoP (*p* < 0.001). To a lesser extent, changes in contents of *ent*-16(*RS*)-9-*epi*-ST-Δ^14^-10-PhytoF, *ent*-9(*RS*)-12-*epi*-ST-Δ^10^-13-PhytoF and 4(*RS*)-4-F_3t_-NeuroP_DPAn-6_ were also statistically supported (*p* < 0.005) (Appendix A). This concerned four of the six families of NEO-PUFAs identified, i.e., those derived from ALA, EPA, DPA_n-6_ and AdA, while the contents of the derivatives from EPA and DHA did not change significantly. Comparing the contents of each of the four families between the control and copper stress condition indicated an increase by 160%, 184%, 104% and 93% for the derivatives of ALA, AA, AdA and for DPA_n-6_, respectively, i.e., a two to three-fold increase in isoprostanoid contents (Figure 4). The concentration of *ent*-9-D_1t_-PhytoP, the metabolite from ALA with the highest content in the control condition (65 ng/g), reached a value of 172 ng/g after cupric stress. Similarly, the content of 5-F_2c_-IsoP increased from 6 ng/g to 20 ng/g, of *ent*-7(*RS*)-F_2t_-dihomo-IsoP from 3 ng/g to 14 ng/g and, of 4(*RS*)-4-F_3t_-NeuroP_DPAn-6_, from 135 ng/g to 260 ng/g. None of the inventoried compounds shown a decrease in contents after applying any of the two oxidative stresses.

### 3.4. Chaetoceros Gracilis

In this diatom, 28 different isoprostanoids were identified, derived from all the PUFAs mentioned above, except DPA_n-6_, and accounted for 2.45 µg/g (Table 4).

Seven phytoprostanoids and phytofuranoides derived from ALA were observed and represented 0.019 µg/g. The concentrations ranged from 1.4 ng/g (*ent*-16-*epi*-16-F_1t_-PhytoP) to 4.4 ng/g (*ent*-16(*RS*)-9-*epi*-ST-Δ^14^-10-PhytoF). Eight neuroprostanoids from DHA (0.077 µg/g), six EPA derivatives (2.2 µg/g), five AA derivatives (0.16 µg/g) and two AdA derivatives (0.011 µg/g) were also identified. DHA derivatives had contents ranging from 3.3 ng/g (10-F_4t_-NeuroP) to 14.6 ng/g (13B(*RS*)-13-F_4t_-NeuroP), while EPA derivatives accumulated from 52.4 ng/g (8-*epi*-8-F_3t_-IsoP) to 535 ng/g (18-*epi*-18-F_3t_-IsoP) and AA derivatives from 9.3 ng/g (15-F_2t_-IsoP) to 105 ng/g (5-F_2c_-IsoP). As it could be expected based on the high content of EPA found in *C. gracilis* [39,43,44], the most abundant isoprostanoids identified in this species were derived from this PUFA, notably the diasteroisomer pair 5(*RS*)-5-F_3t_-IsoP that accounted for approximately 42% (1.1 µg/g) of the total amount of oxidized metabolites measured.

For this alga, the qualitative profile remained mostly unchanged, similarly to the amounts of the individual molecules, between the control and the two oxidative stress conditions tested. The only exception was for the compound 18-F_3t_-IsoP; it showed a slight significant difference in its concentration under the Cu^2+^ stress condition, increasing from 362 ng/g to 428 ng/g (*p* = 0.015) (Figure 5 and Appendix A).

### 3.5. Phaeodactylum Tricornutum

In this diatom, 21 different oxidized metabolites were identified and quantified, for a total of 0.32 µg/g. No derivatives of DPA_n-6_ were observed under any conditions. ALA derivatives represented the main isoprostanoids in term of diversity with 12 metabolites (six PhytoPs and six PhytoFs) and, also, in terms of the content (0.21 µg/g, i.e., 66% of the total amount). Concentrations of ALA-oxidized metabolites were comprised within a range of 2.6 ng/g (with *ent*-16(*RS*)-13-*epi*-ST-Δ^14^-9-PhytoF) to 44.2 ng/g (9-F_1t_-PhytoP). The second-most abundant derivatives were produced from EPA, with four metabolites that accounted for 0.09 µg/g. The dynamic range was from 1.2 ng/g for 8-*epi*-8-F_3t_-IsoP to 47 ng/g for (5(*R*)-5-F_3t_-IsoP. One single isoprostanoid from AA was identified (5-F_2c_-IsoP; 13 ng/g), two from AdA (7(*RS*)-ST-Δ^8^-11-dihomo-IsoF epimers; 5 ng/g) and, finally, two from DHA (4(*RS*)-4-F_4t_-NeuroP epimers; 2 ng/g). No oxidized derivatives of DPA_n-6_ were found, as previously stated for the other diatom, *C. gracilis* (Table 5). As already mentioned for *T. lutea* and *C. gracilis*, *P. tricornutum* produced isoprostanoids in accordance with its PUFA profile that contained mainly ALA and EPA [39,45].

The isoprostanoid profile of the diatom *P. tricornutum* was strongly influenced by the copper treatment, in contrast to what was observed for the other diatom, *C. gracilis* (Figure 6).

A significant increase in the content of 14 metabolites among the 21 identified was observed (Appendix A). This was particularly significant for *ent*-16(*RS*)-13-*epi*-ST-Δ^14^-9-PhytoF (*p* < 0.00005), with a concentration of 2.6 ng/g in the control condition and 7 ng/g under the copper stress condition. Compounds 16-B_1_-PhytoP, 9-L_1_-PhytoP, *ent*-16(*RS*)-9-*epi*-ST-Δ^14^-10-PhytoF and *ent*-9(*RS)*-12-*epi*-ST-Δ^10^-13-PhytoF were also strongly impacted by copper, with significant modifications in the contents (*p* < 0.0005). The concentrations of these different metabolites increased from 15 ng/g to 43 ng/g, 12 ng/g to 34 ng/g, 20 ng/g to 44 ng/g and 4.5 ng/g to 12 ng/g, respectively. The amounts of the derivatives of ALA were enhanced by 94%, of AA by 94%, of AdA by 105% and of DHA by 58%, representing, on average, a two-fold increase in the isoprostanoid contents. No changes in the isoprostanoids for which EPA is the precursor was observed. No metabolite showed a decrease in contents.

Surprisingly, we monitored the lower levels of the isoprostanoids under H_2_O_2_ stress compared to the control condition for *P. tricornutum*. The contents of the two epimer series derived from ALA oxidation, *ent*-16(*RS*)-13-*epi*-ST-Δ^14^-9-PhytoF and *ent*-16(*RS*)-9-*epi*-ST-Δ^14^-10-PhytoF, decreased by factors of two and four, respectively (*p* < 0.005). In the same vein, the amounts of 16-B_1_-PhytoP and 9-L_1_-PhytoP were four times lower under the H_2_O_2_ stress condition compared to the control (*p* < 0.05).

## 4. Discussion

There is an increasing interest in studying oxylipin metabolism in marine microalgae. This is supported by recent publications describing profiles of enzymatically produced oxidized derivatives of PUFAs and their potential physiological roles [17,46,47,48,49]. So far, little emphasis has been put on biosynthesis, by eukaryotic phytoplankton, of isoprostanoids, i.e., oxylipins produced nonenzymatically by a reaction of ROS with the double-bonds of PUFAs. In this context, and to our knowledge, the current study is the first to report the production of phytoprostanes, phytofurans, isoprostanes (serie 2 and 3) and neuroprostanes, all derived from PUFA precursors that include ALA, AdA, EPA, DPA_n-6_ and DHA, in the cryptophyte *R. salina*, the haptophyte *T. lutea* and the diatom *C. gracilis*. In addition, it extends the repertoire of isoprostanoids recently published for another diatom, *P. tricornutun* [19]. Under laboratory culture growth conditions, a good correlation between the presence of PUFAs and the biosynthesis of NEO-PUFAs was observed. We have already noticed this in our previous study on macroalgae, notably with the *Rhodophyta* species *Grateloupia turuturu* Yamada, known to be rich in AA and which produced significant amounts of oxidized AA metabolites [18]. The four microalgae investigated in the present study exhibited different levels of diversity, as well as marked differences in the amounts of isoprostanoids produced. A high content of ALA derivatives was quantified in *R. salina*. The diatom *P. tricornutum*, which appeared to contain lower amounts of oxidized derivatives compared to the other microalgae used, was mostly rich in AA derivatives. The diatoms *C. gracilis* showed profiles rich in NEO-PUFAs produced from EPA. High contents of ALA and DHA derivatives were identified in *T. lutea*. After exposure to oxidative stress conditions, changes in the diversity and amounts of isoprostanoids produced were species and stress-dependent. Under copper stress, no strong variations were observed in *R. salina* and *C. gracilis*, whereas a significant increase in the production of C18-, C20- and C22-derived isoprostanoids was monitored in *T. lutea*. and *P. tricornutum*. H_2_O_2_ stress had different impacts. NEO-PUFA concentrations remained unchanged for *C. gracilis* and *T. lutea*, whereas profiles and contents were altered in *R. salina* and *P. tricornutum*, notably for the ALA-oxidized derivatives. Changes in phytoprostanes derived from ALA have been recently observed in this latter alga under a H_2_O_2_ treatment slightly different from the condition considered in our analysis (1 mM for 24 h), i.e., 0.25 and 0.75 mM of H_2_O_2_ applied during 48 h [19]. Interestingly, this study identified a number of isoprostanoids derived from ALA, ARA, EPA and DHA, which levels were differentially affected after oxidative stress. The authors have studied the influence of nine synthetic isoprostanoids, applied in the micromolar range, on the physiology and lipid metabolism of *P. tricornutum.* They observed an induction of the accumulation of triacylglycerols (storage lipids) and a reduction of growth without the alteration of photosynthesis. Such a study, describing the characterization of nonenzymatic oxylipins in *P. tricornutum* and suggesting physiological roles for these molecules, paved the way to better understand their importance in the biology of marine microalgae.

In the context of research on microalgal biorefinery, numerous studies were conducted on culture parameters and have been shown to impact the production of PUFAs, such as light [50], macronutrient depletion [51,52], temperature [53] or salinity [54]. Nitrogen depletion or salinity stress, for instance, were shown to induce oxidative stress and significant changes in PUFA productions [51], but little is known about the impacts of these abiotic parameters on oxidized PUFA derivatives. Under laboratory culture growth conditions, our study showed that isoprostanoid profiles present good correlations with PUFA contents, and that their production could be increased in *T. lutea*, *P. tricornutum* and *R. salina* by applying direct oxidative stress, either through a copper or H_2_O_2_ addition. According to previous studies on lipid metabolism regulation, these results suggest that culture condition manipulation could also be an interesting field to be explored for improving the biotechnological production of microalgal isoprostanoids.

From a more methodological point of view, it is worth underlining the sensitivity of the measurements, since we have managed to measure metabolites present in very small quantities. Indeed, considering the case of *P. tricornutum* and of its lowest represented isoprostanoid (8-*epi*-8-F_3t_-IsoP) presents at approximately 1 ng/g, it is satisfying to detect and reliably quantify molecules at such very low levels. With LODs ranging between 0.16 ng/g and 0.63 ng/g and LOQs comprised between 0.16 ng/g and 1.25 pg/g, we can consider the method as sensitive. Interestingly, these low detection limits enabled to detect and quantify a great diversity of metabolites. To our knowledge, the 38 oxidized metabolites detected in *T. lutea* represent the highest diversity of isoprostanoids identified from a given organism so far, including plants and animals. This number of 38 isoprostanoids is close to the number of molecules for which we currently have standards for our targeted lipidomics method (47 metabolites). However, we cannot ensure that these undetected compounds were absent. Some of the missing metabolites may be present in amounts lower than our LODs/LODs or may be produced but masked by other molecules from the matrix. To overcome these two issues, it is possible to foresee improving the sample preparation and, also, the chromatographic procedure by working on the choice of column, solvents and gradient. Furthermore, it is important to emphasize that more than 38 isoprostanoids may be present in the microalgae studied but could not be identified, because our analysis is based on a targeted lipidomics approach and, thus, only detects the metabolites present in the analytical method. Therefore, we think it is important to consider extending the library of isoprostanoid/oxylipin standards through the synthesis of new molecules by chemists, as well as to adopt an untargeted lipidomics method [55,56] to expand the investigation of algal isoprostanoids in the future. Another important observation is that there is a very large difference in the concentrations of NEO-PUFAs in the four species investigated, although this can be partly smoothed out by applying the correction factors of the extraction yield and matrix effect.

Finally, recent studies have shown promising biological activities for PhytoPs, IsoPs and NeuroPs [16,57]. For instance, Minghetti et al. showed the ability of the phytoprostane B_1_-PhytoP, through novel mechanisms involving PPAR-γ, to specifically affect immature brain cells, such as neuroblasts and oligodendrocyte progenitors, thereby conferring neuroprotection against oxidant injury and promoting myelination [58]. Duda et al. showed the role, also as lipid mediator, of some phytoprostanes in the immediate effector phase of allergic inflammation [38]. More recently, the work of Lee et al. put forward the hypothesis of the neuroprotective effect of 4-F_4t_-NeuroP in cellular and animal models [59]. Early studies on the cardiovascular system demonstrated that AA-oxidized derivatives induced platelet aggregation or showed a hypertensive effect [60,61]. More recent studies showed that IsoPs and NeuroPs have beneficial effects in cardiovascular disease. Indeed, Leguennec et al. revealed that the lipid mediator 4-F_4t_-NeuroP derived from the nonenzymatic peroxidation of DHA has an antiarrhythmic effect in ventricular cardiomyocytes and in post-myocardial infarcted mice [62]. They also demonstrated the capability of such derivatives to prevent and protect rat myocardium from reperfusion damages following occlusion (ischemia) [63]. Due to high amounts quantified in some of the tested microalgae, especially after copper exposure, it may be worth exploring these organisms as a potential natural resource for the production of isoprostanoids. The extraction of these NEO-PUFAs from marine microalgae could be an interesting alternative to current productions by complex chemical syntheses, as are macroalgae. In this context, further works should focus on assessing how culture conditions alter the isoprostanoid contents and diversity in selected algae towards enhancing productions for future extractions from natural resources.

## 5. Conclusions

The current investigation aimed at profiling isoprostanoids by micro-LC-MS/MS in selected marine microalgae belonging to different lineages: the cryptophyte *Rhodomonas salina*, the haptophyte *Tisochrysis lutea* and the diatoms *Phaeodactylum tricornutum* and *Chaetoceros gracilis*. To our knowledge, this is the first report of such a wide variety of NEO-PUFAs produced in microalgae. For instance, our analysis allowed the detection of PUFA-oxidized derivatives never reported so far, and we detected no less than 38 different metabolites in *T. lutea*. Our study is also the first to establish a link between significant changes in the isoprostanoid profiles of some selected microalgae and heavy metal stress. It also highlights the impact of hydrogen peroxide stress on NEO-PUFAs in some cases. Based on recent studies showing promising biological activities for NEO-PUFAs and due to high amounts quantified in some of the tested microalgae, further works should focus on assessing how manipulating culture conditions could enhance the production of isoprostanoids in selected species, notably by targeting the PUFA biosynthetic pathways.

## Figures and Tables

**Figure 1 biomolecules-10-01073-f001:**
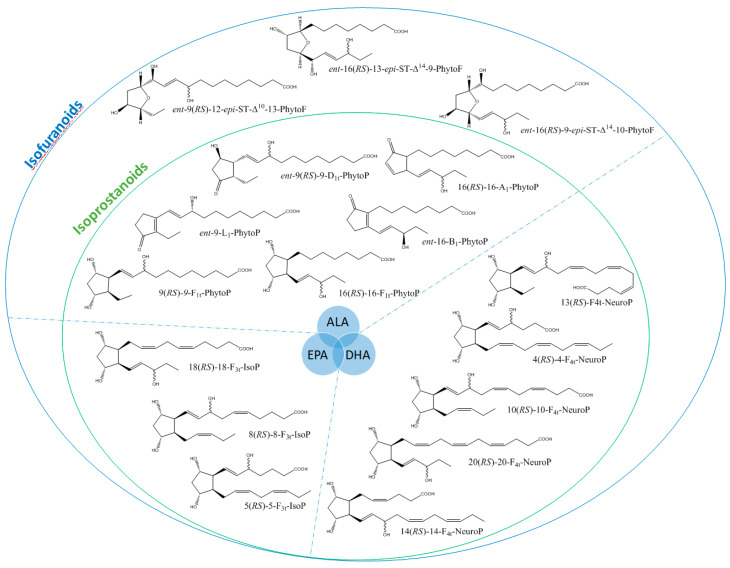
Structure of some isoprostanoids isomers derived from n-3 polyunsaturated fatty acids (PUFAs): ALA (α-linolenic acid), EPA (eicosapentaenoic acid) and DHA (docosahexaenoic acid).

**Figure 2 biomolecules-10-01073-f002:**
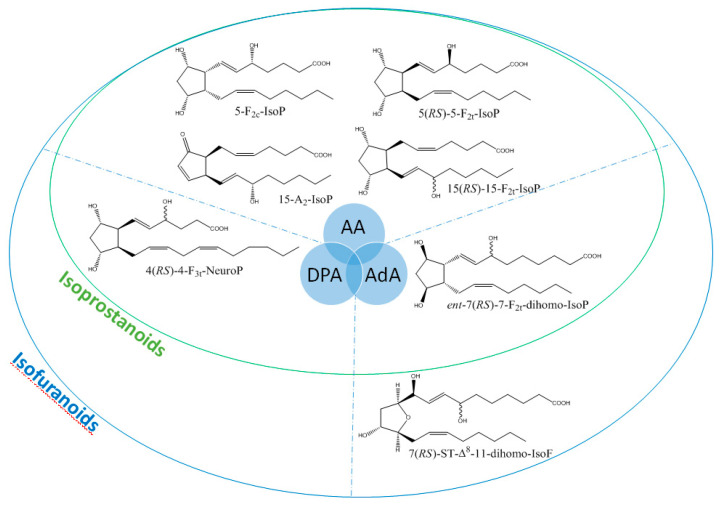
Structure of some isoprostanoids isomers derived from n-6 PUFAs: AA (arachidonic acid), DPA_n-6_ (docosapentaenoic acid) and AdA (adrenic acid).

**Figure 3 biomolecules-10-01073-f003:**
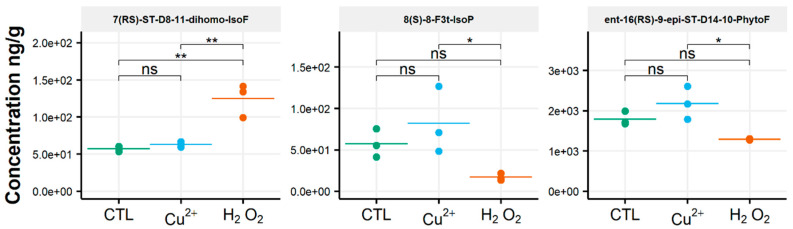
Changes in contents of selected isoprostanoids for the cryptophyte *Rhodomonas salina* between the control condition (CTL) and oxidative stress (Cu^2+^ and H_2_O_2_) conditions. Statistically relevant responses between the control and stress conditions (one-way ANOVA) are indicated by asterisks: * *p* < 5 × 10^−2^ and ** *p* < 5 × 10^−3^; ns, not significant.

**Figure 4 biomolecules-10-01073-f004:**
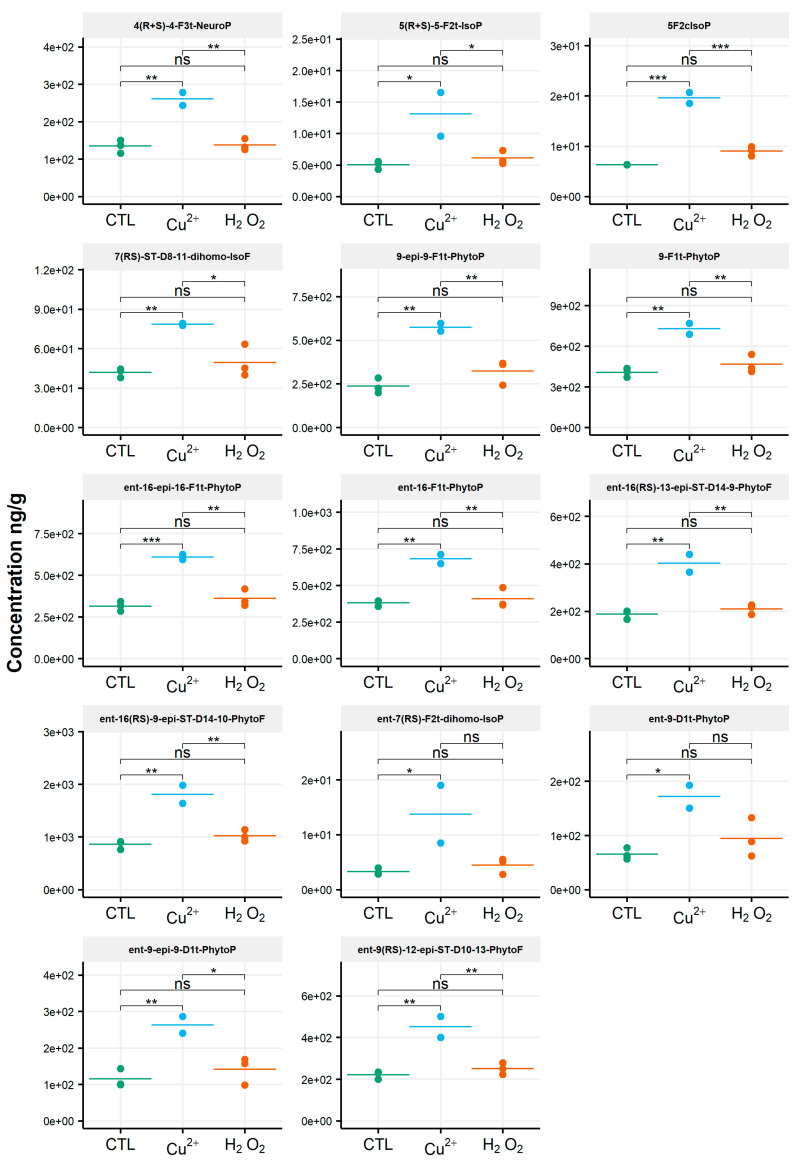
Changes in contents of selected isoprostanoids for the haptophyte *Tisochrysis lutea* between the control condition (CTL) and oxidative stress (Cu^2+^ and H_2_O_2_) conditions. Statistically relevant responses between the control and stress conditions (one-way ANOVA) are indicated by asterisks: * *p* < 5 × 10^−2^, ** *p* < 5 × 10^−3^ and *** *p* < 5 × 10^−4^; ns, not significant.

**Figure 5 biomolecules-10-01073-f005:**
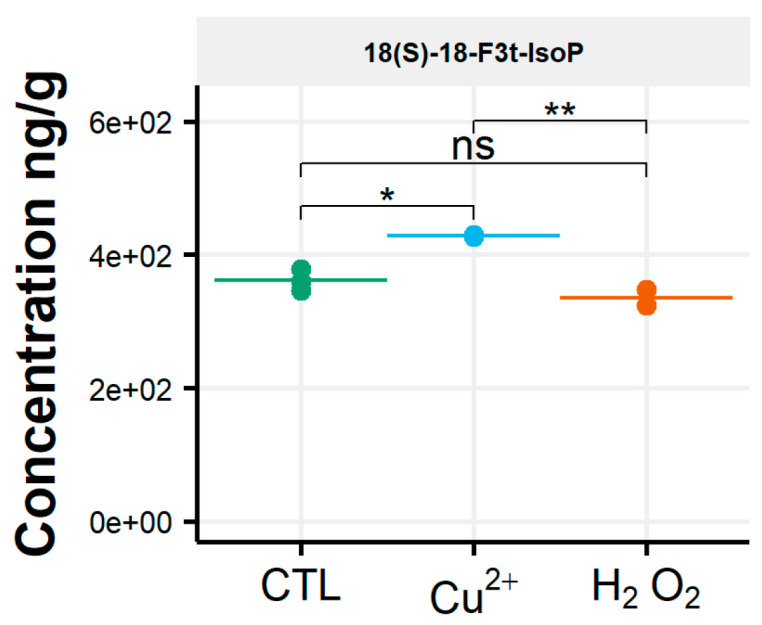
Changes in content of selected isoprostanoids for the diatom *Chaetoceros gracilis* between the control condition (CTL) and oxidative stress (Cu^2+^ and H_2_O_2_) conditions. Statistically relevant responses between the control and stress conditions (one-way ANOVA) are indicated by asterisks: * *p* < 5 × 10^−2^ and ** *p* < 5 × 10^−3^; ns, not significant.

**Figure 6 biomolecules-10-01073-f006:**
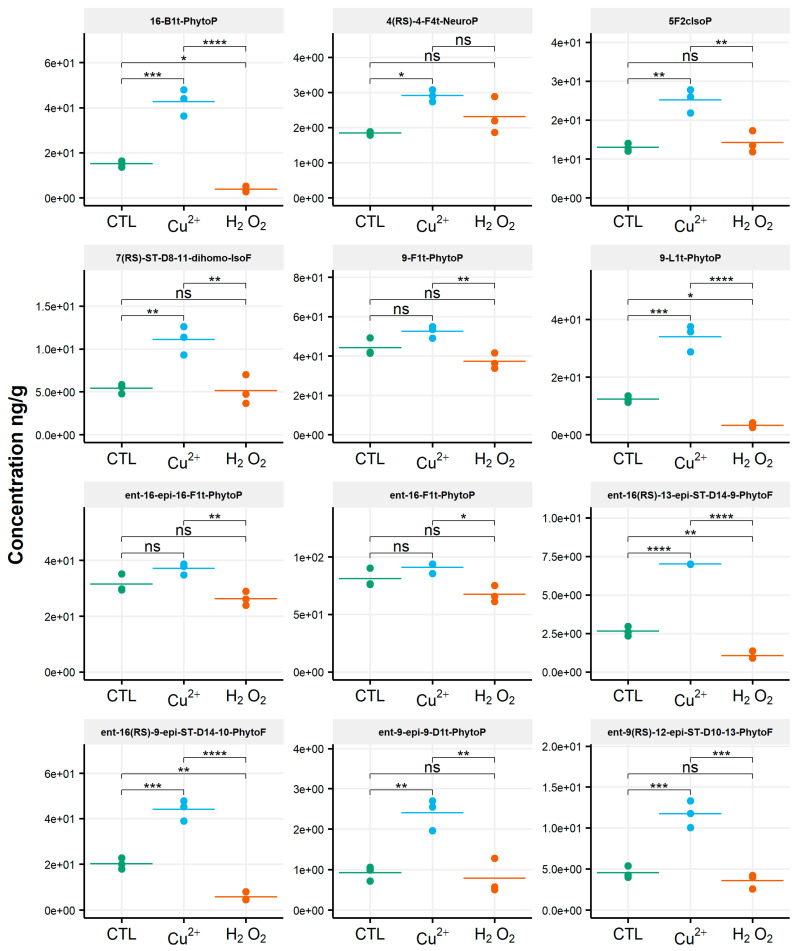
Changes in contents of selected isoprostanoids for the diatom *Phaeodactylum tricornutum* between the control condition (CTL) and oxidative stress (Cu^2+^ and H_2_O_2_) conditions. Statistically relevant responses between the control and stress conditions (one-way ANOVA) are indicated by asterisks: * *p* < 5 × 10^−2^, ** *p* < 5 × 10^−3^, *** *p* < 5 × 10^−4^ and **** *p* < 1 × 10^−4^; ns, not significant.

**Table 1 biomolecules-10-01073-t001:** Relative percentage distribution of each type of nonenzymatic oxidation-polyunsaturated fatty acids (NEO-PUFAs) in *C. gracilis, P. tricornutum, T. lutea* and *R. salina*: ALA (α-linolenic acid), AA (arachidonic acid), AdA (adrenic acid), EPA (eicosapentaenoic acid), DPA_n-6_ (docosapentaenoic acid) and DHA (docosahexaenoic acid).

Microalgal Species	Metabolites of ALA	Metabolites of AA	Metabolites of AdA	Metabolites of EPA	Metabolites of DPA	Metabolites of DHA
*C. gracilis*						
CTL	0.8%	6.6%	0.4%	89.0%	0.0%	3.1%
Cu^2+^	1.1%	7.4%	0.5%	87.9%	0.0%	3.0%
H_2_O_2_	0.8%	11.2%	0.4%	83.4%	0.0%	4.2%
*P. tricornutum*						
CTL	65.5%	4.0%	1.7%	28.2%	0.0%	0.6%
Cu^2+^	58.1%	4.5%	2.0%	34.8%	0.0%	0.5%
H_2_O_2_	44.8%	4.3%	1.5%	48.7%	0.0%	0.7%
*T. lutea*						
CTL	69.5%	0.2%	0.6%	0.1%	1.9%	27.7%
Cu^2+^	73.6%	0.3%	0.8%	0.1%	2.2%	23.0%
H_2_O_2_	67.9%	0.2%	0.7%	0.1%	1.7%	29.4%
*R* *. salina*						
CTL	71.2%	2.4%	0.5%	18.2%	0.1%	7.4%
Cu^2+^	66.7%	2.7%	0.4%	21.4%	0.2%	8.6%
H_2_O_2_	79.1%	3.5%	1.5%	12.4%	0.1%	3.4%

**Table 2 biomolecules-10-01073-t002:** Quantification of NEO-PUFAs in *R. salina* incubated under control, copper or H_2_O_2_ stress conditions. Data are mean ± sd (n = 3) expressed as ng/g dry weight. NaN stands for Not a Number, because it is an impossible value.

Component Name	CTL		Cu^2+^		H_2_O_2_	
	Conc.	sd	Conc.	sd	Conc.	sd
10-*epi*-10-F_4t_-NeuroP	49.6	18.1	81.2	51.5	13.2	5.03
10-F_4t_-NeuroP	40.0	11.9	62.5	34.3	15.1	5.99
13-*epi*-13-F_4t_-NeuroP	113	35.7	160	76.8	39.2	13.4
13-F_4t_-NeuroP	183	65.2	279	153	41.0	NaN
14(*RS*)-14-F_4t_-NeuroP	51.0	14.8	89.0	55.0	12.3	4.78
15-*epi*-15-F_2t_-IsoP	26.0	7.17	37.1	15.4	31.7	9.39
15-F_2t_-IsoP	14.0	3.78	21.9	10.1	17.3	6.42
16-B_1_-PhytoP	1960	96.9	2190	208	1410	844
18-F_3t_-IsoP	711	239	1100	575	343	142
18-*epi*-18-F_3t_-IsoP	240	63.6	393	203	174	54.9
20-*epi*-20-F_4t_-NeuroP	67.0	19.7	97.2	48.0	37.3	16.2
20-F_4t_-NeuroP	88.8	28.7	143.0	81.6	33.1	11.4
4(*RS*)-4-F_3t_-NeuroP	13.4	4.57	22.7	16.8	9.24	1.90
4(*RS*)-4-F_4t_-NeuroP	194	45.0	326	187	90.1	9.40
5-*epi*-5-F_3t_-IsoP	457	137	717	378	278	77.1
5(*RS*)-5-F_2t_-IsoP	70.1	17.7	107	49.0	95.7	22.3
5-F_3t_-IsoP	424	107	713	377	193	43.3
5-F_2c_-IsoP	149	31.0	222	96.2	143	34.9
7(*RS*)-ST-Δ^18^-11-dihomo-IsoF	57.1	3.55	62.9	3.56	125	22.4
8-*epi*-8-F_3t_-IsoP	38.9	11.5	61.4	33,0	18.9	7.81
8-F_3t_-IsoP	57.3	17.0	82.0	40.2	17.0	4.21
9-*epi*-9-F_1t_-PhytoP	514	115	851	455	668	119
9-F_1t_-PhytoP	584	113	888	404	687	122
9-L_1_-PhytoP	1510	76.6	1660	172	1110	619
*ent*-16-*epi*-16-F_1t_-PhytoP	440	96.7	715	362	550	118
*ent*-16-F_1t_-PhytoP	311	66.4	520	284	417	81.3
*ent*-16(*RS*)-9-*epi*-ST-Δ^14^-10-PhytoF	1790	173	2180	411	1290	NaN
*ent*-9(*RS*)-12-*epi*-ST-Δ^10^-13-PhytoF	434	NaN	570	124	404	149

**Table 3 biomolecules-10-01073-t003:** Quantification of NEO-PUFAs in *T. lutea* incubated under control, copper or H_2_O_2_ stress conditions. Data are mean ± sd (n = 3, except for copper stress: n = 2) expressed as ng/g dry weight. NaN stands for Not a Number, because it is an impossible value.

Component Name	CTL		Cu^2+^		H_2_O_2_	
	Conc.	sd	Conc.	sd	Conc.	sd
10-*epi*-10-F_4t_-NeuroP	137	12.4	190	NaN	175	51.7
10-F_4t_-NeuroP	105	10.3	153	NaN	132	37.5
13-*epi*-13-F_4t_-NeuroP	282	15.4	312	NaN	334	71.2
13-F_4t_-NeuroP	433	33.3	505	NaN	543	125
14(*RS*)-14-F_4t_-NeuroP	132	15.8	159	NaN	167	72.9
16-B_1_-PhytoP	988	99.0	1010	NaN	1050	169
16(*RS*)-16-A_1_-PhytoP	324	30.2	587	NaN	348	39.3
18-F_3t_-IsoP	4.54	1.13	4.60	NaN	2.46	NaN
20-*epi*-20-F_4t_-NeuroP	116	15.3	213	NaN	144	33.2
20-F_4t_-NeuroP	203	20.8	326	NaN	256	60.0
4(*RS*)-4-F_3t_-NeuroP	135	17.5	261	NaN	138	15.3
4(*RS*)-4-F_4t-_NeuroP	515	38.7	826	NaN	601	102
5(*RS*)-5-F_2t_-IsoP	5.05	0.635	13.1	NaN	6.10	1.09
5-F_2c_-IsoP	6.33	NaN	19.6	NaN	9.03	0.925
7(*RS*)-ST-Δ^18^-11-dihomo-IsoF	41.8	3.50	78.5	NaN	49.5	12.2
8-*epi*-8-F_3t_-IsoP	1.24	0.344	2.43	NaN	1.90	0.939
8-F_3t_-IsoP	2.27	0.448	4.21	NaN	3.64	0.97
9-*epi*-9-F_1t_-PhytoP	237	43.4	575	NaN	324	71.2
9-F_1t-_PhytoP	407	33.6	730	NaN	466	65.9
9-L_1_-PhytoP	727	82.2	1300	NaN	759	128
*ent*-16-*epi*-16-F_1t_-PhytoP	315	28.8	610	NaN	361	50.7
*ent*-16-F_1t_-PhytoP	381	20.4	682	NaN	409	66.6
*ent*-16(*RS*)-13-epi-Δ^14^-9-PhytoF	188	18.6	402	NaN	210	21.6
*ent*-16(*RS*)-9-*epi*-ST-Δ^14^-10-PhytoF	859	84.5	1810	NaN	1020	107
*ent*-7(*RS*)-7-F_2t_-dihomo-IsoP	3.26	0.602	13.7	NaN	4.47	1.48
*ent*-9-D_1t_-PhytoP	65.5	10.9	172	NaN	94.6	35.6
*ent*-9-*epi*-9-D_1t-_PhytoP	115	24.9	263	NaN	142	38.1
*ent*-9(*RS*)-12-*epi*-ST-Δ^10^-13-PhytoF	221	18.9	451	NaN	251	27.4

**Table 4 biomolecules-10-01073-t004:** Quantification of NEO-PUFAs in *C. gracilis* incubated under control, copper or H_2_O_2_ stress. Data are mean ± sd (n = 3) expressed as ng/g dry weight. NaN stands for Not a Number, because it is an impossible value.

Component Name	CTL		Cu^2+^		H_2_O_2_	
	Conc.	sd	Conc.	sd	Conc.	sd
10-*epi*-10-F_4t_-NeuroP	5.24	0.437	6.23	1.32	5.68	NaN
10-F_4t_-NeuroP	3.28	0.217	3.97	0.503	8.23	8.04
13-*epi*-13-F_4t_-NeuroP	11.0	0.779	14.3	1.51	24.7	22.8
13-F_4t_-NeuroP	14.6	1.88	16.6	3.60	31.8	25.8
15-*epi*-15-F_2t_-IsoP	13.2	0.443	16	0.24	19.3	14.1
15-F_2t_-IsoP	9.32	0.449	10.8	1.24	14.7	11.0
16-B_1_-PhytoP	4.38	0.913	7.04	2.10	4.22	1.29
18-F_3t_-IsoP	635	33.5	767	NaN	1040	792
18-*epi*-18-F_3t_-IsoP	362	16.1	428	NaN	335	NaN
20-*epi*-20-F_4t_-NeuroP	9.38	0.976	10.3	NaN	15.8	11.5
20-F_4t_-NeuroP	10.7	2.15	10.3	2.79	17.7	13.4
4(*RS*)-4-F_4t_-NeuroP	22.7	1.62	23.2	4.11	23.9	NaN
5-*epi*-5-F_3t_-IsoP	603	26.6	661	92.3	569	NaN
5(*RS*)-5-F_2t_-IsoP	35.1	3.19	41.4	7.19	61.1	52.8
5-F_3t-_IsoP	471	22.2	477	80.4	493	NaN
5-F_2c_-IsoP	105	3.71	140	20.1	248	180
7(*RS*)-ST-Δ^18^-11-dihomo-IsoF	10.8	0.754	14.2	0.802	12.9	2.66
8-*epi*-8-F_3t_-IsoP	57.5	3.01	63.7	7.63	56.0	NaN
8-F_3t_-IsoP	52.4	2.05	62.9	7.24	56.6	NaN
9-F_1t-_PhytoP	2.06	0.136	2.43	0.0889	2.83	1.29
9-L_1_-PhytoP	3.21	0.587	5.40	1.53	3.27	0.92
*ent*-16-*epi*-16-F_1t_-PhytoP	1.44	0.164	1.84	0.229	2.04	0.952
*ent*-16(*RS*)-9-*epi*-ST-Δ^14^-10-PhytoF	4.44	0.354	7.08	1.33	5.51	1.59
*ent*-9-*epi*-9-D_1t_-PhytoP	3.96	0.347	6.51	1.02	7.54	5.75

**Table 5 biomolecules-10-01073-t005:** Quantification of NEO-PUFAs in *P. tricornutum* incubated under control, copper or H_2_O_2_ stress conditions. Data are mean ± sd (n = 3) expressed as ng/g dry weight. NaN stands for Not a Number, because they are impossible values.

Component Name	CTL		Cu^2+^		H_2_O_2_	
	Conc.	sd	Conc.	sd	Conc.	sd
16-B_1_-PhytoP	15.1	1.38	42.7	5.94	3.80	1.23
4(*RS*)-4-F_4t_-NeuroP	1.84	0.0457	2.91	0.169	2.31	0.518
5-*epi*-5-F_3t_-IsoP	47.1	2.74	99.9	14.0	86.6	49.4
5-F_3t_-IsoP	33.7	2.07	72.9	8.36	57.9	32.4
5-F_2c_-IsoP	13.0	1.02	25.2	3.05	14.2	2.78
7(*RS*)-ST-Δ^18^-11-dihomo-IsoF	5.41	0.554	11.1	1.66	5.14	1.69
8-*epi*-8-F_3t_-IsoP	6.72	0.759	12.0	1.83	10.8	4.19
8-F_3t_-IsoP	3.92	0.304	7.99	0.911	6.92	3.48
9-F_1t_-PhytoP	44.2	4.36	52.5	3.05	37.2	3.93
9-L_1_-PhytoP	12.3	1.13	34.0	4.63	3.25	0.846
*ent*-16-*epi*-16-F_1t_-PhytoP	31.5	3.19	37.1	2.10	26.3	2.54
*ent*-16-F_1t_-PhytoP	81.0	8.05	90.9	4.69	67.3	7.15
*ent*-16(*RS*)-13-*epi*-Δ^14^-9-PhytoF	2.64	0.312	7.01	NaN	1.06	0.268
*ent*-16(*RS*)-9-*epi*-ST-Δ^14^-10-PhytoF	20.2	2.46	44.0	4.61	5.78	1.94
*ent*-9-*epi*-9-D_1t_-PhytoP	0.918	0.182	2.40	0.392	0.783	0.426
*ent*-9(*RS*)-12-*epi*-ST-Δ^10^-13-PhytoF	4.53	0.744	11.7	1.65	3.58	0.882

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
