# Peer review of "Isoprostanoid Profiling of Marine Microalgae"

_biomolecules, 2020, doi:10.3390/biom10071073_

Round 1

Reviewer 1 Report

See attached. 

Reviewer 2 Report

The study is very interesting and innovative, however, the following corrections must be done:

Figures 1 and 2: Names and structures are too small; it cannot be read.

Line 136: Complete name of species

Line 159-164: Why did you add this information? This indicate that samples were compromised/degraded and should not be used in the assay. The n are not the same? It should be the same among tested conditions.

Line 253-256: This means that the assay was not correct and thus result may not be used to take any conclusions. This should be repeated.

Line 293: Table S4 does not exist, it may be a mistake

All figures except 1 and 2: Graphs should be bigger to a better understanding Added and individual YY’s label at each graph and the units are missing. In the XX’s axis, put the names horizontally. Identify in the Fig. legend the meaning of “ns” and “*”.

All table : information is missing in the units, “µg” of what and “g” of what? Check SD units, maybe a “.” are missing. Also add statistics to show significant differences among conditions.

Reviewer 3 Report

This paper deals with

Aim of this study is

My global comment is that the experiments are well conducted, objectives are suitably described and followed thorough the manuscript and conclusions are in correlation with the experimental results. The content of the paper is in the scope of the journal, the presentation globally complies to the “guide for authors” and Figures and tables are relevant.

I consider then that the manuscript could be published in Biomolecules, after some minor corrections listed below:

-Figure 1 and Figure 2 are too small, it is not possible to clearly see the chemical structures of molecules and to read their names

-line 106: “R. salina, we incubated” replace “we” by “were”

- Lines 107-110: “Isoprostanoid profiles obtained for these microalgae showed good correlation with their content in PUFA precursors. Oxidative stress conditions were found to have different impacts on oxidized derivative contents according to the microalgal species investigated” The end of introduction contains already some results, this is not the good place to put these lines.

- Lines 159-164, materials and methods: “During the preparation of samples for such analysis, we made two important observations. First, we noticed that one sample of P. tricornutum obtained under H2O2 stress condition contained some water after lyophylization. This sample was not considered further for extraction. In addition, one sample of T. lutea obtained after copper stress shown significant difference in color and texture compared to the other samples after starting the extraction process, therefore it was not used for lipidomic profiling.” These lines already correspond to “results” and should not be placed here.

-The above-mentioned lines suggest that freeze-drying conditions were not suitable for this kind of sample, but details are not in the manuscript. Please add

- lines 171-172: “4 μL of IS (1 ng/μL), and 1.5 mL of phosphate buffer (pH 2) saturated in NaCl” What is IS (internal standard? Which compound?)? What is the molarity of the phosphate buffer? What is the counter-ion of phosphate (Na? K?) How can you reach a pH of 2 with phosphate compound?

Round 2

Reviewer 1 Report

Vigor and colleagues made significant enhancements to the current manuscript in their revision, however we fundamentally disagree about the lighting conditions used during the culture of their phytoplankton. I do agree with the authors that culture conditions from the RCC do not necessarily reflect normal culture conditions, yet I am unaware of other studies using culture conditions in these ranges. To address this issue, I suggest that the authors either address the high light conditions in the discussion or to simply add citations to other studies or protocols that culture these specific species at high light conditions.

Author Response

We have just carried out new measurements in the culture room with another quantameter, different from the one used for the first verification. This quantameter also has a spherical probe (Biospherical Instrument QSL2101 and the associated software Logger 2100). The average values observed are 300 µmoles/m2/s as measured previously. We can therefore attest that these light conditions are those applied and that they are conducive to the active growth of biomass of the different strains studied. 

Moreover, a review of the literature also allows us to report that these light conditions have already been applied to other cultures such as Tisochrysis (del Pilar Sanchez-Saavedra, 2016) or Phaeodactylum (Heydarizadeh, 2017). We have added these two references to our manuscript.

I hope this answer will suit you.